# Causal Relationship between Meat Intake and Biological Aging: Evidence from Mendelian Randomization Analysis

**DOI:** 10.3390/nu16152433

**Published:** 2024-07-26

**Authors:** Shupeng Liu, Yinyun Deng, Hui Liu, Zhengzheng Fu, Yinghui Wang, Meijuan Zhou, Zhijun Feng

**Affiliations:** Department of Radiation medicine, Guangdong Provincial Key Laboratory of Tropical Disease Research, School of Public Health, Southern Medical University, Guangzhou 510515, China; 20220062@smu.edu.cn (S.L.); yinyun9845@foxmail.com (Y.D.); lhuidoc@163.com (H.L.); smufuzz@163.com (Z.F.); wyh045@smu.edu.cn (Y.W.)

**Keywords:** causal relationship, meat consumption, biological aging, Mendelian randomization, red meat, white meat, processed meat

## Abstract

Existing research indicates that different types of meat have varying effects on health and aging, but the specific causal relationships remain unclear. This study aimed to explore the causal relationship between different types of meat intake and aging-related phenotypes. This study employed Mendelian randomization (MR) to select genetic variants associated with meat intake from large genomic databases, ensuring the independence and pleiotropy-free nature of these instrumental variables (IVs), and calculated the F-statistic to evaluate the strength of the IVs. The validity of causal estimates was assessed through sensitivity analyses and various MR methods (MR-Egger, weighted median, inverse-variance weighted (IVW), simple mode, and weighted mode), with the MR-Egger regression intercept used to test for pleiotropy bias and Cochran’s Q test employed to evaluate the heterogeneity of the results. The findings reveal a positive causal relationship between meat consumers and DNA methylation PhenoAge acceleration, suggesting that increased meat intake may accelerate the biological aging process. Specifically, lamb intake is found to have a positive causal effect on mitochondrial DNA copy number, while processed meat consumption shows a negative causal effect on telomere length. No significant causal relationships were observed for other types of meat intake. This study highlights the significant impact that processing and cooking methods have on meat’s role in health and aging, enhancing our understanding of how specific types of meat and their preparation affect the aging process, providing a theoretical basis for dietary strategies aimed at delaying aging and enhancing quality of life.

## 1. Introduction

Changes and trends in global meat consumption have had profound impacts on public health and the ecological environment [1]. In recent years, especially in rapidly developing economies, meat consumption levels have significantly increased [2,3]. This trend has sparked widespread attention and research from health scholars regarding its potential health effects [4,5]. Studies indicate that high meat consumption, particularly of red and processed meats, is closely associated with the occurrence of various chronic diseases, including cardiovascular diseases [6,7,8], cancer [9,10,11,12], and metabolic syndrome [13,14,15]. For example, several studies indicated that higher intake of red and processed meats was linked to increased mortality from cardiovascular diseases and cancer [12,16,17]. In contrast, the intake of white meat (such as poultry and fish) is generally associated with lower health risks [18,19,20]. These findings underscore the importance of distinguishing between red and white meat in dietary intake concerning health risks, providing a scientific basis for public health policy formulation.

Aging is a complex biological process involving a series of physiological and functional changes [21,22]. Studying biomarkers of aging-related phenotypes is crucial for understanding the mechanisms of aging and promoting healthy aging [23,24]. Commonly used aging biomarkers include telomere length [25,26], mitochondrial DNA copy number [27,28,29], and DNA methylation levels [30,31,32]. Telomere length is an important marker of cellular aging, shortening with cell division, and is thus used as an indicator of an individual’s biological age [33,34,35]. Changes in mitochondrial DNA copy number are also closely related to the aging process [36,37]. In previous analyses, we identified a negative causal relationship between telomere length shortening and mitochondrial DNA copy number, indicating that as telomeres shorten, aging accelerates, and mitochondrial DNA copy number significantly decreases [38]. Similar results have been reported in other studies [39,40,41,42]. In recent years, with advancements in molecular biology techniques, assessing and predicting biological age through detecting DNA methylation levels at specific sites has become a hotspot in aging research, using methods such as the Horvath clock [43,44,45,46,47,48]. In daily life, aging is often manifested as declines in cognitive function and physical fitness, which significantly affect an individual’s quality of life and independence. The role of diet structure in the aging process has also gradually become a research focus [49,50,51,52]. Meat intake, as a primary source of animal protein in daily life, plays an important role in ensuring healthy life activities, yet its potential health risks still need to be thoroughly explored. Currently, there is no definitive conclusion on whether meat intake causes or accelerates aging-related phenotypes. 

The Mendelian randomization (MR) method uses genetic variations as instrumental variables to study the causal relationship between exposure factors (such as meat consumers) and outcomes (such as aging-related phenotypes) [53,54]. The significant advantage of the MR method is its ability to reduce the impact of confounding biases and reverse causation common in observational studies, thereby providing more reliable causal inferences [55]. Genetic variations are randomly assigned during fertilization, a natural randomization process akin to the random grouping in randomized controlled trials, giving the MR method a unique advantage in causal inference [56,57]. This study used the MR method, selecting genetic variants associated with meat intake as instrumental variables (IVs) to analyze their association with aging-related phenotypes. Through large-sample data analysis, we aimed to clarify the mechanisms by which meat consumption affects the aging process and verify the causal relationship between different types of meat intake and aging-related phenotypes. Specifically, this study aimed to identify genetic variants associated with meat consumers and use them as IVs; analyze the causal relationship between these genetic variants and various aging-related phenotypes (such as telomere length, mitochondrial DNA copy number, DNA methylation levels related to human age); infer the causal relationship between meat consumers and aging-related phenotypes through the MR method; and explore the different impacts of red, white, and processed meat intakes on the aging process. Through these research steps, we aimed to reveal the causal relationship of meat intake on the aging process, provide the public with more scientific dietary recommendations, and promote healthy aging.

## 2. Material and Methods

### 2.1. Study Design and Data Source

This study followed the STROBE-MR guidelines (Strengthening the Reporting of Observational Studies in Epidemiology Using Mendelian Randomization) [58] and employed a two-sample MR approach. Eleven GWAS datasets related to meat intake were retrieved and downloaded from the openGWAS database (https://gwas.mrcieu.ac.uk/, accessed on 10 June 2024). These datasets include one for overall meat consumers [59], three for red meat (including pork, lamb, and beef) intake, three for white meat (including poultry, oily fish, and non-oily fish) intake [60], and four for processed meat (including overall processed meat consumption, ham, bacon, and sausage) intake. Additionally, four aging-related phenotype datasets were selected as outcome variables, including telomere length [61], mitochondrial DNA copy number [62], PhenoAge (a DNA methylation predictor trained on 42 clinical markers and age), and Hannum age (intrinsic epigenetic age acceleration) [46]. Detailed information on these datasets is provided in Appendix A. Based on these data, a comprehensive MR analysis framework (Figure 1) was constructed to explore the potential causal relationships between meat intake and aging-related phenotypes.

To ensure the robustness of the MR analysis, three key assumptions regarding IVs were established [63,64]: (1) IVs must be significantly associated with the exposure; (2) IVs must not be related to any confounders that might simultaneously affect the exposure and the outcome; (3) IVs must influence the outcome solely through the exposure and not via any other pathways. Since all data used in this study were obtained from publicly available databases, no additional ethical approval or informed consent was required. Figure 1 provides a detailed description of the dataset selection, IV screening, and main steps of the MR analysis.

### 2.2. IVs Selection and Data Cleaning

According to Figure 1, the “TwoSampleMR” R package was utilized to obtain IVs related to meat consumers and meat intake [65], ensuring sufficient sample sizes and appropriate effect allele frequencies (EAFs). Missing EAFs were supplemented using data from the 1000 Genomes Project (https://www.internationalgenome.org/home, accessed on 10 June 2024) [66]. Only single nucleotide polymorphisms (SNPs) with an F-statistic (F = β^2^/se^2^) greater than 10 were included in the analysis [67,68]. Data cleaning involved removing potential confounding IVs identified through the “LDtrait” database [69]. Additionally, IVs with *p*-values less than 0.05 were considered outliers and excluded using the “RadialMR” R package (https://github.com/WSpiller/RadialMR/, accessed on 10 June 2024) [70], with further assessment of IV pleiotropy conducted via Global *t*-tests [71,72].

### 2.3. MR Analysis

Five MR analysis methods (MR-Egger [73], weighted median [74], inverse-variance weighted (IVW) [75], simple mode [76], and weighted mode [77]) were employed to comprehensively assess the impact of meat consumers and different types of meat intake on aging-related phenotypes [75,78,79]. Causality was judged based on two criteria: (1) consistent direction of causal effect estimates (*β* values or *B* values) across various methods; (2) statistical significance primarily reliant on a *P*_ivw_ value of less than 0.05. Each direction of analysis was conducted independently, and no multiple comparison corrections were applied, with significance determined using the original *p*-values.

### 2.4. Sensitivity Analysis

Sensitivity analyses included heterogeneity analysis and pleiotropy testing, performed using Cochran’s Q test and the MR-Egger method [80,81]. The MR-Egger intercept test was used to evaluate horizontal pleiotropy [82], with a significance level set at *p* < 0.05. For IVW analysis, a fixed-effect model was used in the absence of heterogeneity, whereas a random-effect model was applied when heterogeneity was present (*P_Q test_* < 0.05) [83]. Additionally, to clarify the independent causal effects of each IV on the outcome variable in various analytical directions, we assessed the causal effect of individual SNP. We also conducted a leave-one-out analysis to observe the causal effect on the outcome variable after sequentially removing each SNP. Both sets of results are presented using forest plots. Funnel plots and scatter plots were utilized for visual assessments of heterogeneity and pleiotropy in the MR analyses. 

## 3. Results

### 3.1. Causal Relationship between Meat Consumers and Aging-Related Phenotypes

According to MR guidelines, 27 SNPs significantly associated with meat consumers (ukb-b-12276) were identified, with an average F-statistic of 24.64 (Appendix A). The MR analysis results (Table 1) revealed a significant positive causal effect between meat consumers and DNA methylation PhenoAge acceleration (all five MR methods yielded positive *B* values, and *P*_IVW_ = 0.02). However, no significant causal effects were observed between meat consumers and telomere length, mitochondrial DNA copy number, or DNA methylation Hannum age acceleration (*P*_IVW_ > 0.05). These results suggest that meat consumers may have a more direct impact on certain specific aging biomarkers, such as DNA methylation PhenoAge acceleration, while its effects on other biomarkers might be limited. Despite the lack of significant associations with telomere length and mitochondrial DNA copy number, the potential complex and indirect impacts of meat consumers on these indicators cannot be ruled out. These findings highlight the differential effects of meat consumers on various aging phenotypes, suggesting that dietary guidelines should carefully consider the health effects related to total meat intake to delay biological aging and reduce the risk of age-related diseases.

### 3.2. Sensitivity Analysis

To ensure the reliability and robustness of the causal relationship between meat consumers and aging-related phenotypes, sensitivity analyses were conducted. The results of pleiotropy and heterogeneity analyses (Table 2) indicated no significant heterogeneity or pleiotropy in the MR analyses of meat consumers and the four aging-related phenotypes (*p* > 0.05). Evaluating the causal relationships from the perspective of individual SNPs showed diverse causal effects on different aging-related phenotypes (Appendix A). This diversity further supports the complex role of meat consumers in biological aging. The leave-one-out analysis confirmed that, excluding any single SNP, the remaining SNPs maintained consistent causal effects on the outcome variables in the analyses of meat consumers with telomere length (Figure 2A), mitochondrial DNA copy number (Figure 2B), and DNA methylation PhenoAge acceleration (Figure 2C). However, in the analysis of meat consumers and DNA methylation Hannum age acceleration (Figure 2D), excluding rs11690184 reversed the causal effect of the remaining SNPs, suggesting potential heterogeneity in this direction. Combining individual SNP effect estimates and leave-one-out analysis results confirmed that some SNPs exhibited broad and differential effects on certain aging biomarkers, but the overall causal effect remained consistent.

Scatter plots depicted the potential associations between the causal effects of the instrumental variables on the exposure and the outcome. The results (Figure 3) showed no significant slope changes in the relationships between meat consumers and telomere length (Figure 3A) or mitochondrial DNA copy number (Figure 3B), confirming the absence of significant causal relationships. For aging-related phenotypes, the SNP effect estimates showed no significant slope changes in the relationship between meat consumers and DNA methylation Hannum age acceleration (Figure 3C), confirming no significant causal relationship, whereas a significant causal relationship was observed with DNA methylation PhenoAge acceleration (Figure 3D). Funnel plots for each analysis direction (Appendix A) did not show obvious abnormal distributions of SNPs. In conclusion, the sensitivity analysis results confirmed that the causal relationships observed in the MR analysis remained consistent across different analytical methods, providing robust support for our conclusion that meat consumption influences the aging process, with multifactorial impacts and significant individual-level differences.

### 3.3. Causal Relationship between Red Meat Intake and Aging-Related Phenotypes

Following MR guidelines, SNPs closely associated with red meat intake were identified, with 168 SNPs for pork intake (average *F*-statistic = 22.82), 209 SNPs for beef intake (average *F*-statistic = 23.04), and 281 SNPs for lamb intake (average *F*-statistic = 23.66) (Appendix A). The results (Figure 4) of MR analysis demonstrated a significant positive causal relationship between lamb intake and mitochondrial DNA copy number (*P*_IVW_ = 0.02). However, no significant results were observed in other analysis directions. Detailed MR analysis results for each direction are provided in Appendix A. This finding highlights a potential important link between lamb intake and mitochondrial function, suggesting dietary habits’ potential impact on mitochondrial health.

### 3.4. Causal Relationship between White Meat Intake and Aging-Related Phenotypes

SNPs closely associated with white meat intake were identified, with 145 SNPs for poultry intake (average *F*-statistic = 22.26), 268 SNPs for oily fish intake (average *F*-statistic = 27.42), and 124 SNPs for non-oily fish intake (average *F*-statistic = 23.89) (Appendix A). The results (Figure 5) showed no significant causal effects between poultry, oily fish, and non-oily fish intake and the four different aging phenotypes (*P*_IVW_ > 0.05). Detailed MR analysis results for each direction are provided in Appendix A. This finding suggests that although these foods are considered potentially beneficial for health, their impact on the aging process may not be as significant as expected, prompting further investigation into the specific effects and biological mechanisms of different food categories on aging.

### 3.5. Causal Relationship between Processed Meat Intake and Aging-Related Phenotypes

SNPs closely associated with processed meat consumption were identified, with 85 SNPs for overall processed meat intake (average *F*-statistic = 23.10), 45 SNPs for ham intake (average *F*-statistic = 20.02), 44 SNPs for bacon intake (average *F*-statistic = 20.89), and 49 SNPs for sausage intake (average *F*-statistic = 20.19) (Appendix A). The MR analysis results (Table 3) confirmed a significant inverse causal relationship between processed meat intake and telomere length (Table 3, *P*_IVW_ < 0.05), but no significant causal effects were observed for the other three aging-related phenotypes (Table 3, *P*_IVW_ > 0.05). In subgroup analyses of processed meats (ham, bacon, sausage intake), no significant causal effects on the four aging-related phenotypes were observed (Figure 6). Detailed MR analysis results for each direction are provided in Appendix A. This evidence indicates that while single types of processed meat may not significantly impact the aging process, the cumulative effect of overall processed meat consumption should not be ignored. These findings emphasize the importance of controlling total processed meat intake and suggest dietary adjustments to reduce processed meat consumption, advocating for a more balanced and healthy diet to prevent health risks associated with processed meat intake.

### 3.6. Heterogeneity and Pleiotropy in Subgroup Analyses

In the sensitivity analyses of subgroups, no significant pleiotropy (Appendix A, *p* > 0.05) or heterogeneity (Appendix A, *p* > 0.05) was observed between different types of meat intake and the four aging-related phenotypes. This stage focused on the distribution of instrumental variables that showed significance in the MR analysis to ensure the reliability and validity of the MR results. In the MR analysis of lamb intake and mitochondrial DNA copy number, although individual SNPs showed differential causal effects on mitochondrial DNA copy number (Appendix A), the remaining SNPs continued to show significant causal effects after excluding any single SNP, consistently positioned to the right of the null line (Appendix A). Furthermore, the scatter plot indicated a significant positive trend in the causal relationship between exposure-related SNPs and the outcome variable (IVW method fitting curve showing a distinct positive slope, Figure 7A), with no significant abnormal SNP distribution observed in the funnel plot (Figure 7B). Similarly, in the MR analysis of overall processed meat intake and telomere length, individual SNPs displayed differential causal effects on telomere length (Appendix A), but the remaining SNPs maintained an overall consistent causal effect after excluding any single SNP, positioned to the left of the null line (Appendix A). The scatter plot showed a significant negative trend in the causal relationship between exposure-related SNPs and the outcome variable (IVW method fitting curve showing a distinct negative slope, Figure 7C), with no significant abnormal SNP distribution observed in the funnel plot (Figure 7D). These results not only confirm the reliability and robustness of the MR analysis but also enhance the credibility of these findings, further supporting their value in guiding dietary recommendations.

## 4. Discussion

This study delves into the causal relationship between different types of meat intake and aging phenotypes (such as telomere length, mitochondrial DNA copy number, and DNA methylation levels related to human age). The results first revealed a positive causal relationship between meat consumption and accelerated DNA methylation aging in meat consumers, suggesting that increased meat intake may accelerate the biological aging process. This finding sets the premise for our subsequent analyses. To further evaluate the causal effects of different types of meat consumption on aging phenotypes, we conducted subgroup analyses based on three categories: red meat, white meat, and processed meat. For red meat, we selected the most commonly consumed types in daily life, namely beef, lamb, and pork. Mendelian randomization (MR) analysis confirmed a positive causal effect of lamb intake on mitochondrial DNA copy number. However, no significant effects were observed for beef and pork on the four aging-related phenotypes. For white meat, we chose the most common types of meat in daily diets: chicken and fish. MR analysis showed no significant association between the types of white meat included in the study (chicken, oily fish, and non-oily fish) and the four aging phenotypes. Processed meat, which includes commonly consumed types such as ham, sausage, and bacon, was also analyzed. MR analysis indicated a negative causal relationship between overall processed meat consumption and telomere length, suggesting that higher intake of processed meat may shorten telomere length. However, no significant associations were found between processed meat consumption and the other three aging phenotypes (mitochondrial DNA copy number and DNA methylation levels related to human age). Subgroup analyses for processed meat also did not reveal any significant causal links between consumption levels and the four aging-related phenotypes. These findings emphasize that the total consumption of processed meat may impact the aging process mediated by telomere length, while the cumulative effect of meat consumption in general may be more significant than the impact of individual types of meat. Therefore, although specific types of meat consumption have varying effects on aging markers, the overall quantity and frequency of meat intake may be more crucial in influencing biological aging. In summary, this study provides new insights into the relationship between different types of meat intake and aging phenotypes, offering scientific evidence for public health recommendations. It emphasizes the importance of choosing healthy meat products and highlights that the potential harm associated with meat consumption is closely linked to cooking methods and the degree of processing. We advise the public to pay closer attention to meat selection and consumption methods to promote healthy aging. This distinction is crucial in guiding more nuanced and effective public health strategies.

The primary advantage of the MR method is its ability to overcome confounding factors and reverse causation issues common in traditional observational studies [84]. Selecting appropriate IVs is crucial for ensuring the validity and reliability of MR analysis [85]. In this study, genetic variants strongly associated with meat intake were selected from large genomic databases, ensuring their independence and pleiotropy-free nature. The *F*-statistic was calculated to evaluate the strength of the IVs, ensuring sufficient statistical power for reliable causal inference. Sensitivity analyses were conducted to validate the robustness of the MR analysis, using various MR methods to assess the validity of causal estimates [81]. Pleiotropy bias was evaluated using the MR-Egger regression intercept test [86], and heterogeneity was assessed using Cochran’s Q test to ensure consistency among the IVs [87]. Sensitivity analysis results showed no significant issues, confirming the robustness of the main analysis results and the reliability of the study conclusions. Overall, this study adheres to MR analysis standards, employing rigorous IV screening, quality control, and sensitivity analyses to investigate the impact of meat intake on aging phenotypes. These findings enhance our understanding of the causal relationship between meat intake and aging, providing a theoretical basis for strategies to delay aging.

Despite establishing causal relationships between certain meat intakes and the aging process through MR analysis, the specific biological mechanisms involved require further investigation. Recent studies suggest that meat consumption significantly impacts human metabolic processes, which may play a crucial role in aging. For instance, red and processed meats contain saturated fatty acids and cholesterol, which are linked to various metabolic diseases [88,89,90]. Meat consumption influences aging through two primary pathways: first, diets with high meat content affect gut microbiota composition and function, indirectly impacting metabolic status and aging [91,92,93,94]. Reduced gut microbiota diversity and increased pathogenic bacteria can induce low-grade chronic inflammation, accelerating aging through various mechanisms [95,96,97,98,99]. Short-chain fatty acids (SCFAs) produced by gut microbiota are vital for metabolic health, and high-meat diets significantly reduce SCFA production, affecting energy metabolism and immune function [100,101,102]. Second, the different nutritional contents of meats can influence aging-related phenotypes by regulating key metabolic pathways. Red meat intake increases *N*-nitroso compounds, associated with cancer and other age-related diseases [103,104]. Additionally, red meat’s L-carnitine and its metabolite trimethylamine-*N*-oxide (TMAO) are linked to increased cardiovascular disease and mortality [105,106,107]. These metabolites impact cardiovascular health and may accelerate aging through oxidative stress and inflammation pathways [108]. Further research indicates that meat intake affects insulin signaling pathways, altering glucose and lipid metabolism, thus impacting the aging process [109,110]. High-meat diets are associated with insulin resistance and a higher incidence of type 2 diabetes, which are considered significant drivers of aging [111,112,113,114,115,116,117]. This study found a significant inverse causal relationship between overall processed meat consumption and telomere length, possibly due to high levels of additives and preservatives in processed meats that increase oxidative stress and cellular DNA damage, leading to accelerated telomere shortening [118,119,120,121,122,123,124]. High saturated fat and salt content in processed meats may also induce chronic inflammation [125,126,127], a critical factor in telomere shortening [21,128,129,130]. These findings highlight the potential negative health impacts of processed meat consumption, emphasizing the importance of reducing processed meat intake and increasing antioxidant and anti-inflammatory foods to delay aging.

In our analysis, a causal relationship was found between overall meat intake and health risks, but subgroup analyses did not show that specific types of meat significantly increased health risks, except for lamb intake, which increased mitochondrial DNA copy number. Mitochondrial DNA copy number is essential for maintaining mitochondrial function and is closely linked to mitochondrial energy production [131]. This result underscores lamb intake’s potential positive impact on mitochondrial health. Several factors may explain this causal relationship: lamb is a rich source of various nutrients [132,133,134,135], including iron, protein, B vitamins (especially B12 and riboflavin), and minerals like selenium, which are vital for normal cell metabolism and biosynthesis. Iron is a necessary cofactor in the mitochondrial electron transport chain, while vitamin B12 is involved in mitochondrial DNA synthesis and maintenance [136]. Specific fatty acids and proteins in lamb might stimulate mitochondrial biogenesis, enhancing energy metabolism efficiency [137,138,139]. Additionally, the antioxidant components in lamb (e.g., selenium) may reduce oxidative stress on mitochondria, protecting mitochondrial DNA and increasing its stability and copy number. The anti-inflammatory properties of certain compounds in lamb might also reduce chronic inflammation, protecting mitochondria and maintaining higher DNA copy numbers [140]. Overall, dietary factors’ regulatory relationships with health are complex, but our evidence supports lamb’s potential benefits in maintaining mitochondrial function and metabolic health. In conclusion, meat intake has multiple effects on metabolism and aging. Future research should focus on uncovering the underlying mechanisms to provide scientific evidence for dietary guidelines and aging intervention strategies. 

This study’s findings, derived from MR analysis, reveal causal relationships between increased meat intake and individual aging risks, offering significant guidance for public health policy and clinical practice. First, the results emphasize the importance of dietary structure in healthy aging, suggesting that public health policies encourage reducing red and processed meat intake while increasing alternatives like fish and poultry. These alternatives are not only nutritious but also lower in saturated fat and cholesterol, helping to reduce the risk of chronic diseases associated with aging. Second, the findings indicate that clinicians should consider potential health impacts from the increase in meat intake when devising personalized dietary plans. For elderly and chronic disease patients, reducing red and processed meat intake and increasing dietary fiber and antioxidant intake can help slow the aging process and improve overall health. This dietary adjustment can improve metabolic status and reduce inflammation, aiding in delaying aging. Third, lamb’s potential in increasing mitochondrial DNA copy number suggests it could be a valuable part of healthy dietary plans, particularly due to its unique nutritional components like conjugated linoleic acid (CLA) and omega-3 fatty acids, which enhance mitochondrial function and replication efficiency. Therefore, dietary recommendations for specific populations might include lamb to optimize energy production and cellular function. 

In summary, this study offers a comprehensive perspective on the nutritional value and health impacts of different types of meat, supporting the formulation of more targeted and effective dietary recommendations. These findings can help optimize individual health and provide scientific evidence for public health policies, promoting healthy aging across society. However, there are limitations to this study. First, MR analysis relies on the reasonableness and independence of genetic variants; some variants may influence multiple phenotypes, causing bias. In addition, there may also be differences in the results produced by both MR analysis methods and statistical methods, such as whether or not to utilize *p*-value corrections. Therefore, the findings of this study still require further validation with real-world data. Second, the study samples mainly come from Western populations, which may limit the applicability of the results to other ethnic groups and regions. Future research should include more diverse samples. Third, due to the complexity of dietary habits and lifestyles, it is challenging to completely exclude confounding factors. Fourth, this study primarily focuses on the impact of different types of meat intake on aging-related phenotypes and does not encompass the diversity of other dietary habits, which may limit our understanding of the overall effects of diet. Future research should combine long-term longitudinal data and various statistical models to further verify the robustness of the causal relationships. Fifth, this study categorizes both fish and poultry under the same classification of “white meat”. This approach does not account for the distinct lipid profiles associated with fish consumption, which differ significantly from those of poultry. Fish is rich in omega-3 fatty acids, which have different health implications compared to the predominantly protein-based profile of poultry. This distinction could affect the associations with aging phenotypes observed in our study. Hence, subsequent studies need to focus on multidisciplinary approaches, utilizing modern biotechnological methods to explore the complex relationship between meat consumption and aging, especially the specific impacts of different types of meat. This will provide theoretical support and practical strategies for achieving healthy aging.

## 5. Conclusions

MR analysis has demonstrated a positive causal effect between meat consumers and accelerated DNA methylation PhenoAge, suggesting that increased meat intake may expedite the biological aging process. Additionally, lamb intake appears to positively influence mtDNA copy number, potentially supporting mitochondrial health and energy metabolism, while processed meats negatively affect telomere length, underscoring the risks associated with their consumption. These variations highlight the significant impact that processing and cooking methods have on meat’s role in health and aging. Consequently, we recommend reducing the intake of processed meats and, where possible, opting for minimally processed alternatives that are cooked in a manner preserving nutritional integrity and reducing harmful effects. This study enhances our understanding of how specific types of meat and their preparation affect the aging process, providing a theoretical basis for dietary strategies aimed at delaying aging and enhancing quality of life.

## Figures and Tables

**Figure 1 nutrients-16-02433-f001:**
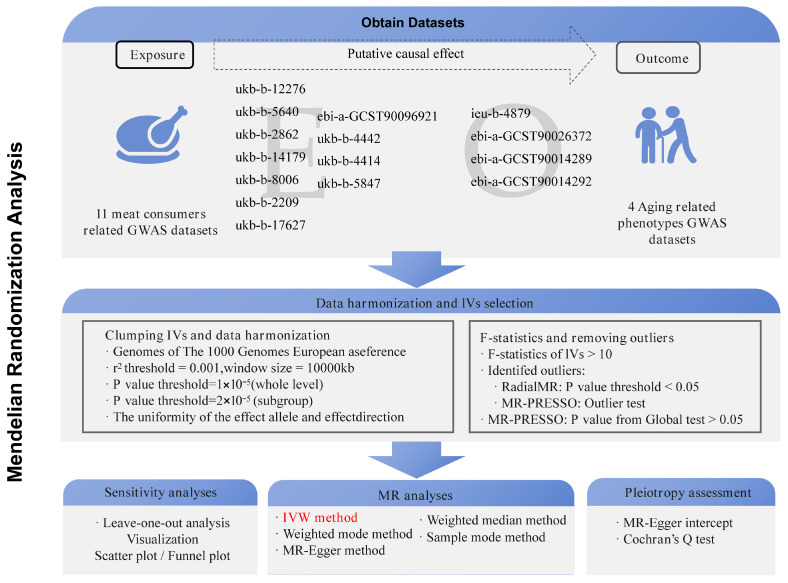
Flow chart showing the design of the study and the steps involved in the data analysis. E represents exposure variable, and the text above it represents the identification of genome-wide association study (GWAS) datasets; O represents outcome variable, and the text above it represents the identification of genome-wide association study (GWAS) datasets; MR, Mendelian randomization; IVW, inverse-variance weighted.

**Figure 2 nutrients-16-02433-f002:**
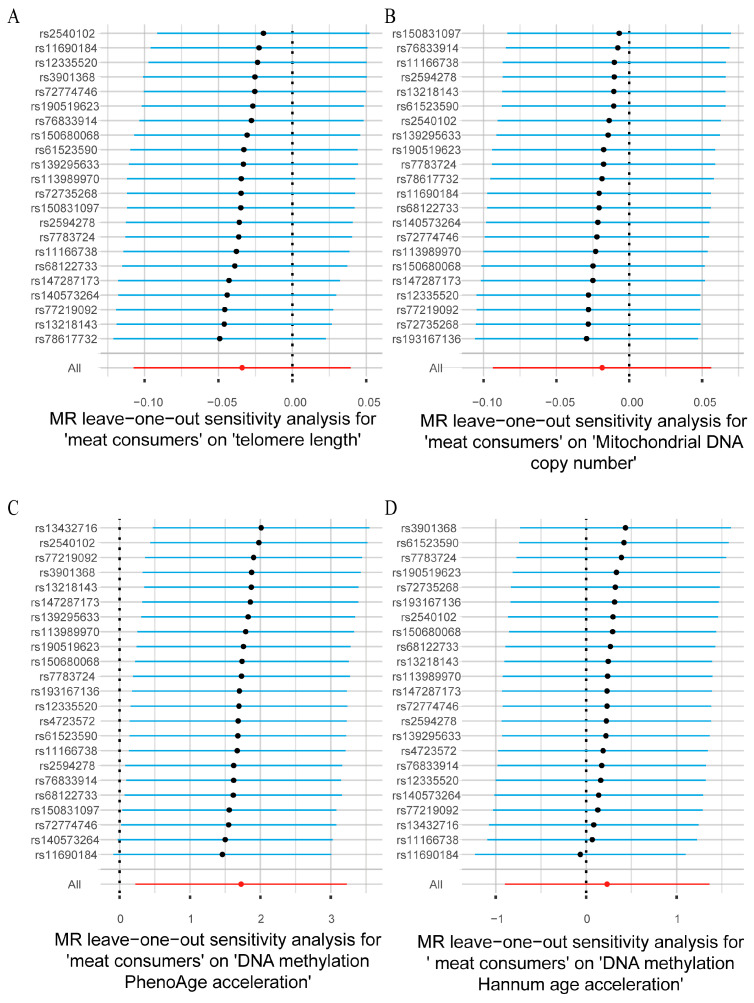
Leave-one-out plots of the causal association between meat consumers and aging-related phenotypes ((**A**) telomere length; (**B**) mitochondrial DNA copy number; (**C**) DNA methylation PhenoAge acceleration; (**D**) DNA methylation Hannum age acceleration). In the plot, each black point represents the recalculated pooled effect size when the corresponding single nucleotide polymorphism is excluded from the analysis. The blue horizontal line represents the maximum (right end) and minimum (left end) estimated values of causal effects. Red points represent cumulative estimates of causal effects, while red horizontal lines represent maximum (right end) and minimum (left end) levels of causal effects.

**Figure 3 nutrients-16-02433-f003:**
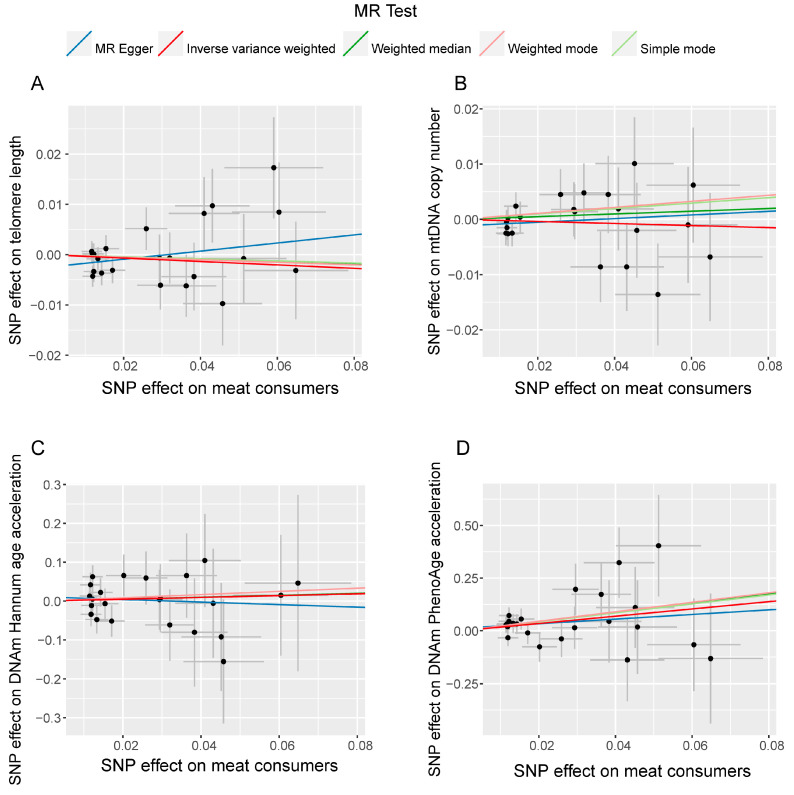
Scatter plots of SNP effects on the meat consumers and the various aging-related phenotypes ((**A**) telomere length; (**B**) mitochondrial DNA copy number; (**C**) DNA methylation PhenoAge acceleration; (**D**) DNA methylation Hannum age acceleration). The X-axis represents the causal effect of single nucleotide polymorphisms (SNPs) on exposure; the Y-axis represents the causal effect of SNPs on outcome; the dots represent SNP for causal estimation in the analysis direction; the different colored lines represent different MR analysis methods as shown in the legend.

**Figure 4 nutrients-16-02433-f004:**
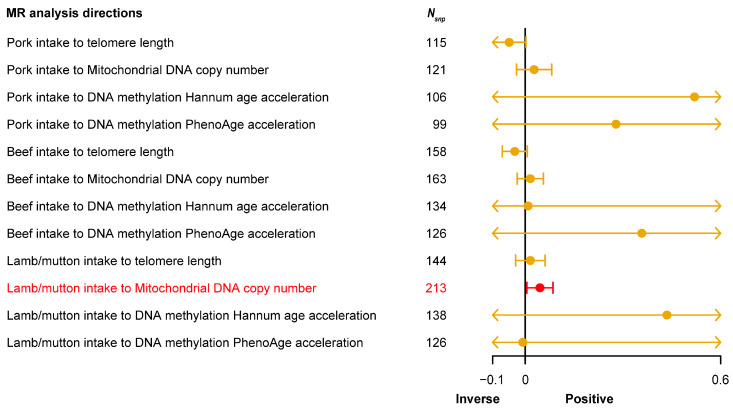
Forest plot shows the result of Mendelian randomization (MR) analysis between red meat (pork, beef, and lamb) intake and four types of aging-related phenotypes. The X-axis represents the direction of causal effect. The vertical line at the 0 scale represents an invalid line. The yellow point in each yellow line represents the causal effect value obtained by the inverse-variance weighted (IVW) method in the current analysis direction, and the yellow horizontal line represents the maximum (right endpoint) and minimum (left endpoint) causal effects estimated by the IVW method in the current analysis direction. If the maximum and minimum values are within the range marked on the X-axis, the end is represented by a small vertical line. When the maximum or minimum value exceeds the range marked on the X-axis, the end is indicated by an arrow. The red section emphasizes that the analysis direction is statistically significant. *N*_snp_, number of single nucleotide polymorphisms (SNPs) in the analysis direction.

**Figure 5 nutrients-16-02433-f005:**
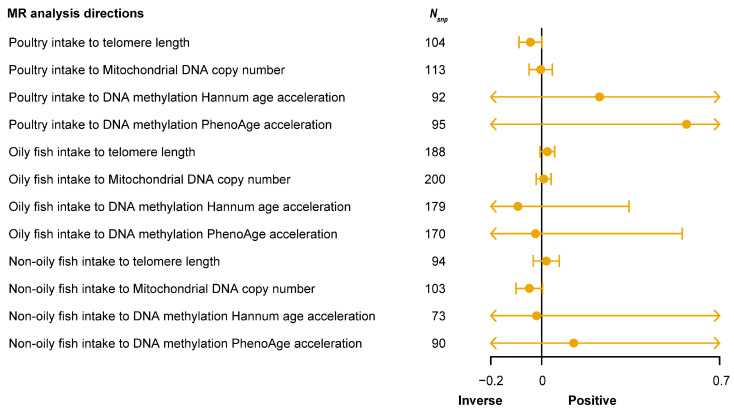
Forest plot shows the result of Mendelian randomization (MR) analysis between white meat (poultry, oily fish, and non-oily fish) intake and four types of aging-related phenotypes. The X-axis represents the direction of causal effect. The vertical line at the 0 scale represents an invalid line. The yellow point in each yellow line represents the causal effect value obtained by the inverse-variance weighted (IVW) method in the current analysis direction, and the yellow horizontal line represents the maximum (right endpoint) and minimum (left endpoint) causal effects estimated by the IVW method in the current analysis direction. If the maximum and minimum values are within the range marked on the X-axis, the end is represented by a small vertical line. When the maximum or minimum value exceeds the range marked on the X-axis, the end is indicated by an arrow. *N*_snp_, number of single nucleotide polymorphisms (SNPs) in the analysis direction.

**Figure 6 nutrients-16-02433-f006:**
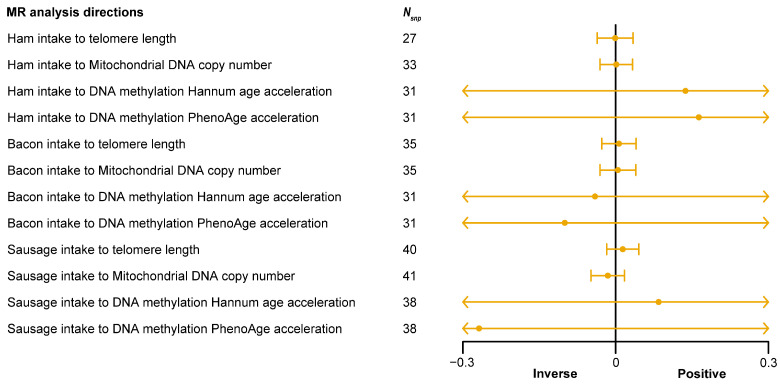
Forest plot shows the result of Mendelian randomization (MR) analysis between processed meat (ham, bacon, and sausage) intake and four types of aging-related phenotypes. The X-axis represents the direction of causal effect. The vertical line at the 0 scale represents an invalid line. The yellow point in each yellow line represents the causal effect value obtained by the inverse-variance weighted (IVW) method in the current analysis direction, and the yellow horizontal line represents the maximum (right endpoint) and minimum (left endpoint) causal effects estimated by the IVW method in the current analysis direction. If the maximum and minimum values are within the range marked on the X-axis, the end is represented by a small vertical line. When the maximum or minimum value exceeds the range marked on the X-axis, the end is indicated by an arrow. *N*_snp_, number of single nucleotide polymorphisms (SNPs) in the analysis direction.

**Figure 7 nutrients-16-02433-f007:**
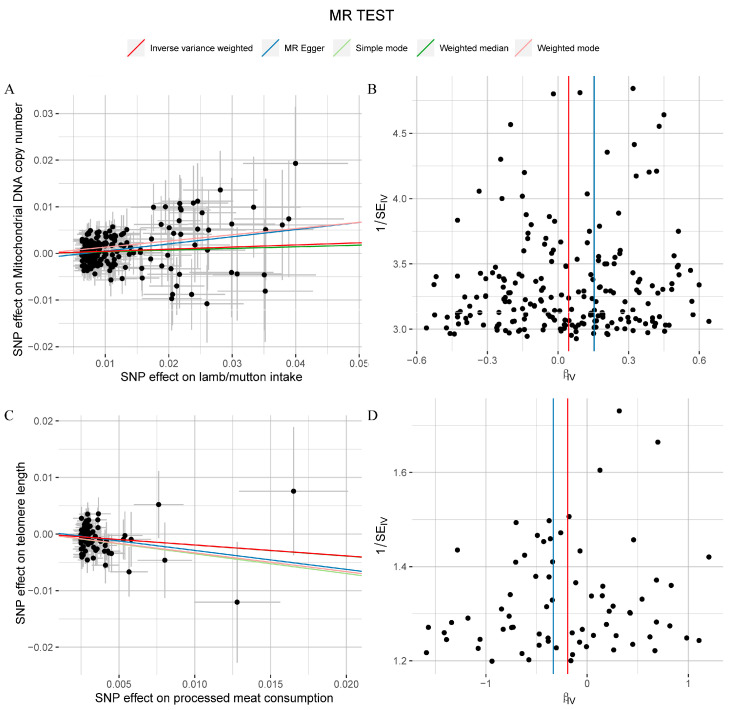
The scatter plots and funnel plots for the Mendelian randomization (MR) analysis between lamb intake and mitochondrial DNA copy number (**A**,**B**), between processed meat consumption and telomere length (**C**,**D**). For scatter plots (**A**,**C**), the X-axis represents the causal effect of single nucleotide polymorphisms (SNPs) on exposure; the Y-axis represents the causal effect of SNPs on outcome; the dots represent SNP for causal estimation in the analysis direction; the different colored lines represent different MR analysis methods as shown in the legend. For funnel plots (**B**,**D**), the X-axis represents the causal effect of single nucleotide polymorphisms (SNPs) on exposure; the Y-axis represents the ratio of 1/standard error (SE) of SNPs from the exposure variable; the dots represent SNP for causal estimation in the analysis direction; the different colored lines represent different MR analysis methods as shown in the legend.

**Table 1 nutrients-16-02433-t001:** The results of Mendelian randomization (MR) analysis between meat consumers and aging-related phenotypes.

Directions and Methods	*N* _snp_	*B*	*SE*	*p*-Value
Meat consumers to telomere length				
MR-Egger	22	0.08	0.07	0.28
Weighted median	22	−0.02	0.05	0.67
Inverse-variance weighted	22	−0.03	0.04	0.36
Simple mode	22	−0.02	0.11	0.85
Weighted mode	22	−0.03	0.10	0.79
Meat consumers to mitochondrial DNA copy number				
MR-Egger	22	0.03	0.07	0.66
Weighted median	22	0.02	0.05	0.63
Inverse-variance weighted	22	−0.02	0.04	0.63
Simple mode	22	0.05	0.10	0.63
Weighted mode	22	0.05	0.10	0.61
Meat consumers to DNA methylation Hannum age acceleration				
MR-Egger	23	−0.33	1.33	0.81
Weighted median	23	0.24	0.82	0.77
Inverse-variance weighted	23	0.23	0.58	0.69
Simple mode	23	0.41	1.67	0.81
Weighted mode	23	0.41	1.62	0.80
Meat consumers to DNA methylation PhenoAge acceleration				
MR-Egger	23	1.10	1.79	0.55
Weighted median	23	2.20	1.07	0.04
Inverse-variance weighted	23	1.73	0.77	0.02 *
Simple mode	23	2.12	2.12	0.33
Weighted mode	23	2.25	1.93	0.26

* statistical significance; *N*_snp_, number of single nucleotide polymorphisms (SNPs) in the MR analysis; *B*, the causal effects estimated by different MR methods; *SE*, standard error.

**Table 2 nutrients-16-02433-t002:** The results of pleiotropy and heterogeneity test in the Mendelian randomization (MR) analysis between meat consumers (exposure) and different aging-related phenotypes (outcomes).

Outcomes	Pleiotropy			Heterogeneity		
	Intercept *	SE	*p*-Value	*Q* Test	*Q*_df	*p*-Value
Telomere length	−0.003	0.001	0.083	22.953	21	0.347
mtDNA copy number	−0.001	0.001	0.431	14.716	21	0.837
DNAm Hannum age acceleration	0.011	0.022	0.644	17.088	22	0.759
DNAm PhenoAge acceleration	0.012	0.030	0.701	17.166	22	0.754

* egger_intercept, mtDNA, mitochondrial DNA, DNAm, DNA methylation.

**Table 3 nutrients-16-02433-t003:** The results of Mendelian randomization (MR) analysis between processed meat consumers and aging-related phenotypes.

Directions and Methods	*N* _snp_	*B*	*SE*	*p*-Value
Processed meat consumers to telomere length				
MR-Egger	71	−0.33	0.37	0.37
Weighted median	71	−0.19	0.12	0.12
Inverse-variance weighted	71	−0.19	0.09	0.03 *
Simple mode	71	−0.35	0.33	0.30
Weighted mode	71	−0.33	0.31	0.28
Processed meat consumers to mitochondrial DNA copy number				
MR-Egger	76	−0.11	0.40	0.78
Weighted median	76	−0.15	0.13	0.26
Inverse-variance weighted	76	0.01	0.09	0.89
Simple mode	76	−0.20	0.35	0.56
Weighted mode	76	−0.21	0.34	0.54
Processed meat consumers to DNA methylation Hannum age acceleration				
MR-Egger	67	−0.63	7.63	0.94
Weighted median	67	1.44	1.88	0.44
Inverse-variance weighted	67	1.41	1.38	0.31
Simple mode	67	−4.23	4.79	0.38
Weighted mode	67	−3.91	4.48	0.39
Processed meat consumers to DNA methylation PhenoAge acceleration				
MR-Egger	64	1.25	9.43	0.89
Weighted median	64	2.32	2.46	0.35
Inverse-variance weighted	64	0.97	1.83	0.59
Simple mode	64	4.78	5.99	0.43
Weighted mode	64	5.75	6.05	0.35

* statistical significance; *N*_snp_, Number of Single nucleotide polymorphisms (SNPs) in the MR analysis; *B*, the causal effects estimated by different MR methods; *SE*, standard error.

## Data Availability

All data used in this study are available in the public repository. The code involved in the data analysis process can be obtained by contacting the corresponding author.

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
