# Peer review of "Causal Relationship between Meat Intake and Biological Aging: Evidence from Mendelian Randomization Analysis"

_nutrients, 2024, doi:10.3390/nu16152433_

Round 1

Reviewer 1 Report

Comments and Suggestions for Authors

The authors propose a topic that could be of great interest, but the method used and the premises considered contain significant gaps:

- The term meat consumption is too generic; it should be characterized with precise quantities, also related to sex, age, and level of physical activity.

- Fish and poultry are grouped in the same group, which is incorrect.

- The cooking methods, which are decisive for the final effect, are not considered.

- It is stated, citing few works, that meat consumption harms health.

- The rest of the eating habits that can negatively and positively influence the overall outcome are not analyzed

Long life should also be related to maintaining muscle mass (surviving is not enough; living healthy should be the goal!).

- We talk about microbiota without analyzing anything that concerns it directly, and in any case, the correlations have to be proven

Comments on the Quality of English Language

It should be revised

Author Response

Dear Professor,

Thank you very much for your thorough review of our manuscript titled "Causal Relationship between Meat Intake and Biological Aging: Evidence from Mendelian Randomization Analysis". We greatly appreciate your insightful comments and suggestions, which have been invaluable in improving the quality of our work. We have carefully considered each of your points and have addressed them as follows:

Comment 1: The term meat consumption is too generic; it should be characterized with precise quantities, also related to sex, age, and level of physical activity.

Response: Thank you very much for your professional advice. As you pointed out, "the term meat consumption is too generic". Following your guidance, we revisited the dataset names in the public database and confirmed that the term used is "meat consumers" (Figure S1 under below and Table S1 in manuscript, ukb-b-12276, https://gwas.mrcieu.ac.uk/datasets/ukb-b-12276/). Initially, we assumed that "meat consumption" and "meat consumers" conveyed the same meaning, and now realize this may have led to a potential misdirection in our approach.

Figure s1 the datasets used in this MR analysis

Based on your suggestion, after discussion within our team, we decided to use the dataset's original term "meat consumers" in our paper to more accurately describe the specific population we studied and the relationship between their dietary habits and aging phenotypes. We appreciate your meticulousness and have revised all references from "meat consumption" to "meat intake" in our manuscript, which have been marked for your review.

Additionally, regarding your comments on specifying meat quantities and considering sex, age, and physical activity levels, we revisited the original data literature. The data uploader had adjusted for these demographic variables when compiling the GWAS data. Most GWAS datasets used for MR analysis are composed of such summary data. Due to limitations in data access, we currently can only use the provided summary data and do not have access to more detailed individual information or other clinical features. We will strive to improve and conduct more detailed analyses based on demographic baseline characteristics in our future research.

We look forward to your further review and valuable feedback.

Comment 2: Fish and poultry are grouped in the same group, which is incorrect.

Response: Thank you for your professional advice. Strictly speaking, there are clear differences between these two categories of meat. Our article adopts the classification method for red meat, white meat, and processed meat from previously published articles (Figure s1). Meats from mammals such as beef, lamb, and pork are categorized as red meat, while poultry like chicken and duck, along with seafood such as fish, shrimp, and shellfish, are classified as white meat. Consequently, although these fundamentally different types of meat were grouped together in the analysis, the entire analytical process was conducted independently for each category. We simply chose to report them under the category of white meat in our manuscript. Thank you once again for your comments, and we look forward to your further review.

Comment 3: The cooking methods, which are decisive for the final effect, are not considered.

Response: Thank you for your thorough review. We apologize for the oversight in our manuscript. Upon re-evaluation of the datasets involved in the manuscript, we confirmed that no data regarding meat processing methods were included. Additionally, there was a misunderstanding in our team where "processed meat" was incorrectly assumed to refer to different processing methods, leading to inappropriate mentions of "processing methods" throughout the text. We appreciate your meticulousness and patience. In fact, "processed meat" refers to a category of meat. We have removed all references to "processing methods" from the manuscript and look forward to your further review.

Comment 4: It is stated, citing few works, that meat consumption harms health.

Response: We appreciate your comment but find it somewhat unclear. Regarding the impact of meat consumption on health, citations 4-17 in our manuscript already focus on the effects of meat intake on health. As a reviewer and expert in this field, if you have any additional relevant literature to recommend, we would be happy to include it in our revisions. Thank you once again for your professional input.

Comment 5: The rest of the eating habits that can negatively and positively influence the overall outcome are not analyzed.

Response: Thank you for your meticulous and professional advice. Indeed, our analysis overlooked this aspect, focusing solely on the impact of meat consumption on aging phenotypes, a limitation which arises from the independence of this approach. Following your suggestion, we have added this point to the section on study limitations, as follows: "Fourth, this study primarily focuses on the impact of different types of meat intake on aging-related phenotypes and does not encompass the diversity of other dietary habits, which may limit our understanding of the overall effects of diet." We look forward to your further review and feedback.

Comment 6: Long life should also be related to maintaining muscle mass (surviving is not enough; living healthy should be the goal!).

Response: We truly value your suggestion, which not only enhances the focus of our manuscript by rooting it in practical life experiences but also provides invaluable guidance for our future analyses based on lifestyle habits. We agree that data analysis closely linked to everyday life should be assessed from a holistic perspective, rather than being independently evaluated based solely on statistical results in relation to specific phenotypes of life activities. Thank you once again for your professional input. In response to your comment, we have added a summary at the end of the first paragraph in the Discussion section, highlighting the contribution to health and aging as follows: "It emphasizes the importance of choosing healthy meat products and reducing processed meat intake, advising the public to pay closer attention to meat selection and consumption to promote healthy aging and overall health." We look forward to your review and expert evaluation.

Comment 7: We talk about microbiota without analyzing anything that concerns it directly, and in any case, the correlations have to be proven.

Response: Thank you for your meticulous and rigorous review. We have removed the content related to the gut microbiome in the revised manuscript as follows: "Additionally, modulating gut microbiota can partially counteract the adverse health impacts of meat consumption. Probiotic and prebiotic supplementation is believed to restore gut microbiota balance and reduce inflammation, potentially mitigating the metabolic disturbances and aging induced by meat consumption. Such interventions might become crucial components of future dietary guidelines, particularly for individuals with high meat diets."

We believe that the revisions we have made have significantly improved the manuscript. We are confident that our responses address all the concerns raised and hope that the revised manuscript meets your expectations.

Thank you once again for your time and effort in reviewing our manuscript. We look forward to your further comments and hope for a positive evaluation.

Sincerely,

Zhijun Feng
South Medical University
fengzhj18@lzu.edu.cn/fengzhj18@sina.com

Reviewer 2 Report

Comments and Suggestions for Authors

Dear author,

Thank you for sharing your research.

Comments:
1.  Could you please provide a summary of your study model in a flowchart to include in the manuscript?

2. The main question addressed by the research is whether there is a causal relationship between meat consumption and aging-related phenotypes, and how different types of meat may impact specific aging biomarkers.

3. The study's originality lies in its use of a two-sample Mendelian Randomization (MR) approach to explore the causal effects of meat consumption on various aging biomarkers. It addresses the gap in understanding the differential impacts of different types of meat on aging, an area not extensively covered in existing research.

4. This study adds to the subject area by providing evidence of specific causal relationships between different types of meat and aging biomarkers. It highlights the potential for lamb to positively influence mitochondrial DNA copy number and suggests an inverse relationship between processed meat consumption and telomere length, offering a more nuanced view of how meat consumption affects aging.

5.  Regarding further controls should be considered, the authors should consider incorporating a more diverse sample population to enhance the generalizability of the findings. Additionally, further controls such as long-term longitudinal studies and detailed dietary assessments could help strengthen the causal inferences and account for potential confounding factors.

 6. The conclusions are consistent with the evidence and arguments presented in the study. The main questions regarding the causal relationships between meat consumption and specific aging biomarkers were addressed using the two-sample MR approach. The study successfully highlighted the differential impacts of various meat types, although it acknowledged that further research is needed to fully understand the underlying mechanisms.

7. The references are appropriate and relevant to the study. They provide a solid foundation for the research by citing key studies related to meat consumption, aging biomarkers, and the Mendelian Randomization methodology.

Author Response

Dear Professor,

Thank you very much for your thorough review of our manuscript titled "Causal Relationship between Meat Intake and Biological Aging: Evidence from Mendelian Randomization Analysis". We greatly appreciate your insightful comments and suggestions, which have been invaluable in improving the quality of our work. We have carefully considered each of your points and have addressed them as follows:

Comment 1: Could you please provide a summary of your study model in a flowchart to include in the manuscript?

Response: Thank you very much for your professional advice. Figure 1 in the manuscript is   

We look forward to your further review and valuable feedback.

Comment 2: The main question addressed by the research is whether there is a causal relationship between meat consumption and aging-related phenotypes, and how different types of meat may impact specific aging biomarkers.

Response: Thank you for summarizing the main focus of our research. We appreciate your clear articulation of our study's objective to explore the causal relationship between meat consumption and aging-related phenotypes, and how different types of meat may influence specific aging biomarkers. Your understanding of the core questions we are addressing reaffirms the clarity with which we have presented our aims and methods. We are grateful for your insights and look forward to any further suggestions you may have.

Comment 3:  The study's originality lies in its use of a two-sample Mendelian Randomization (MR) approach to explore the causal effects of meat consumption on various aging biomarkers. It addresses the gap in understanding the differential impacts of different types of meat on aging, an area not extensively covered in existing research.

Response: Thank you for acknowledging the originality and methodological approach of our study. We are pleased that you recognize our use of a two-sample Mendelian Randomization (MR) framework to investigate the causal effects of meat consumption on aging biomarkers. Your appreciation of our efforts to address the gap in understanding how different types of meat differentially impact aging is encouraging. We believe that our work contributes valuable insights to the field, and we are grateful for your positive evaluation of its scope and significance.

Comment 4: This study adds to the subject area by providing evidence of specific causal relationships between different types of meat and aging biomarkers. It highlights the potential for lamb to positively influence mitochondrial DNA copy number and suggests an inverse relationship between processed meat consumption and telomere length, offering a more nuanced view of how meat consumption affects aging.

Response: Thank you for your detailed recognition of the significant contributions our study makes to the field. We appreciate your emphasis on how our findings provide evidence of specific causal relationships between different types of meat and aging biomarkers. Your acknowledgment of our novel insights into the potential positive impact of lamb on mitochondrial DNA copy number, and the inverse relationship between processed meat consumption and telomere length, reinforces the importance of our research. We are grateful for your understanding of the nuanced view our study offers on how meat consumption affects aging.

Comment 5:  Regarding further controls should be considered, the authors should consider incorporating a more diverse sample population to enhance the generalizability of the findings. Additionally, further controls such as long-term longitudinal studies and detailed dietary assessments could help strengthen the causal inferences and account for potential confounding factors.

Response: Thank you for your valuable suggestions on enhancing the generalizability and robustness of our findings. We agree that incorporating a more diverse sample population would indeed broaden the applicability of our results across different demographics. Furthermore, your recommendation to integrate long-term longitudinal studies and detailed dietary assessments into our research approach is well-received. These additions would undoubtedly strengthen the causal inferences and help mitigate potential confounding factors. We mentioned these points into our limitation and will consider these suggestions for future studies to improve the quality and impact of our research. Thank you once again for your constructive feedback.

Comment 6: The conclusions are consistent with the evidence and arguments presented in the study. The main questions regarding the causal relationships between meat consumption and specific aging biomarkers were addressed using the two-sample MR approach. The study successfully highlighted the differential impacts of various meat types, although it acknowledged that further research is needed to fully understand the underlying mechanisms.

Response: Thank you for affirming that the conclusions of our study are consistent with the evidence and arguments presented. We appreciate your acknowledgment of how effectively the two-sample Mendelian Randomization approach addressed the main questions concerning the causal relationships between meat consumption and specific aging biomarkers. We are also grateful that you recognized our efforts to highlight the differential impacts of various types of meat. As you noted, we agree that further research is necessary to elucidate the underlying mechanisms fully. Your feedback is invaluable as it not only supports our current findings but also guides our future research directions. Thank you once again for your thoughtful and constructive review.

Comment 7: The references are appropriate and relevant to the study. They provide a solid foundation for the research by citing key studies related to meat consumption, aging biomarkers, and the Mendelian Randomization methodology.

Response: Thank you for recognizing the relevance and appropriateness of the references we have chosen for our study. Your feedback affirms our efforts in ensuring that our study is well-grounded in the existing literature, and we appreciate your positive evaluation. Thank you once again for your thoughtful comments.

We believe that the revisions we have made have significantly improved the manuscript. We are confident that our responses address all the concerns raised and hope that the revised manuscript meets your expectations.

Thank you once again for your time and effort in reviewing our manuscript. We look forward to your further comments and hope for a positive evaluation.

Sincerely,

Zhijun Feng
South Medical University
fengzhj18@lzu.edu.cn/fengzhj18@sina.com

Reviewer 3 Report

Comments and Suggestions for Authors

The authors conducted a Mendelian Randomized (MR) analysis to examine the causal relationship between meat consumption and aging-related phenotypes. They showed that meat consumption was positively causally related to DNA methylation PhenoAge acceleration, particularly lamb consumption positively affected mitochondrial DNA copy number, while processed meat consumption negatively affected telomere length. This paper needs to be revised in several respects. 

1. the authors need to clarify the limitations of their study. Please explain in more detail the limitations of the current study, especially with regard to data source bias, sample size issues, and limitations of the MR technique used.

2. the current sensitivity analysis may be sufficient, but consider adding a more detailed sensitivity analysis to demonstrate consistency of results across different MR methods.

3. more clearly compare the impact of different red meat, white meat, and processed meat and analyze in detail the impact of each meat type on specific aging biomarkers.

4. introduce adjustments for multiple comparisons, e.g. Bonferroni correction, to reinforce the statistical significance of the results. Also consider subgroup analysis to further examine the impact of different genetic variants. 

5. the authors may wish to consider biological mechanisms for the results obtained from this paper. For example, please further discuss the specific biological mechanisms of how meat consumption affects certain aging-related phenotypes. In particular, please elaborate on the effects of meat consumption on DNA methylation and mitochondrial DNA copy number, citing existing literature.

6. the term PhenoAge is not common and needs to be explained.

Comments on the Quality of English Language

No

Author Response

Dear Professor,

Thank you very much for your thorough review of our manuscript titled "Causal Relationship between Meat Intake and Biological Aging: Evidence from Mendelian Randomization Analysis". We greatly appreciate your insightful comments and suggestions, which have been invaluable in improving the quality of our work. We have carefully considered each of your points and have addressed them as follows:

Comment 1: the authors need to clarify the limitations of their study. Please explain in more detail the limitations of the current study, especially with regard to data source bias, sample size issues, and limitations of the MR technique used.

Response: Thank you for your professional advice. Following your suggestion, we have revised the final paragraph of the Discussion section in our manuscript to address the limitations of the study. We specifically mentioned the impact of the sample sources, population structure, and other factors on the results of this study, as follows: “First, MR analysis relies on the reasonableness and independence of genetic variants; some variants may influence multiple phenotypes, causing bias. Additionally, different MR analysis methods may yield varying results; therefore, the findings of this study still require further validation with real-world data. Second, the study samples mainly come from Western populations, which may limit the applicability of the results to other ethnic groups and regions. Future research should include more diverse samples. Third, due to the complexity of dietary habits and lifestyles, it is challenging to completely exclude confounding factors. Fourth, this study primarily focuses on the impact of different types of meat intake on aging related phenotypes and does not encompass the diversity of other dietary habits, which may limit our understanding of the overall effects of diet.” 

We look forward to your further review.

Comment 2: the current sensitivity analysis may be sufficient, but consider adding a more detailed sensitivity analysis to demonstrate consistency of results across different MR methods.

Response: Thank you for your professional advice. To ensure the reliability and robustness of our MR analysis results, we conducted a strict quality control process on the instrumental variables. Using the "RadialMR" R package, we identified and removed instrumental variables (IVs) with P<0.05 as outliers. We then assessed the reliability and robustness of the IVs post-outlier removal using Global tests. These methodologies have been added and described in the manuscript (LINES 121-123). Additionally, we evaluated the causal impact of individual SNPs on the outcome variables within each analytic direction, and used a leave-one-out analysis to observe the effects of the remaining SNPs on the outcome variables after sequentially removing each SNP. Both sets of results are displayed using forest plots (Lines 138-142). Please review again.

Comment 3: more clearly compare the impact of different red meat, white meat, and processed meat and analyze in detail the impact of each meat type on specific aging biomarkers.

Response: Thank you for your professional advice. Following your suggestions, we have revised the first paragraph of the Discussion section to better summarize the results of the entire paper. We have systematically summarized the findings according to the three categories: red meat, white meat, and processed meat, detailing their respective causal impacts on the outcome variables (Lines 253-278). The manuscript has been resubmitted for your review, and we look forward to your further guidance and assistance.

Comment 4: introduce adjustments for multiple comparisons, e.g. Bonferroni correction, to reinforce the statistical significance of the results. Also consider subgroup analysis to further examine the impact of different genetic variants. 

Response: Thank you for your rigorous and professional suggestions. Initially, we considered applying a Bonferroni correction to the obtained P-values to strengthen the evidence. However, the corrected P-values showed no significance. Consequently, after team discussions, we decided to analyze each analytic direction independently, from extracting exposure-related instrumental variables to outcome-related ones, to optimize the analysis. Additionally, we consulted other studies and confirmed that P-value adjustments might not be necessary for exploratory analyses aimed at uncovering all potential factors that could impact health. Therefore, in this study, we did not adjust the P-values but conducted subgroup analyses for different types of meat. However, given that Mendelian randomization analysis requires consistency within the same ethnicity, and aging-related phenotype data are primarily from European populations, there is still a lack of evidence supporting analyses in other ethnic groups. Following your advice, we have reported the absence of P-value adjustments as a limitation of this study (Lines 373-375). The revised manuscript has been resubmitted for your further review.

Comment 5: the authors may wish to consider biological mechanisms for the results obtained from this paper. For example, please further discuss the specific biological mechanisms of how meat consumption affects certain aging-related phenotypes. In particular, please elaborate on the effects of meat consumption on DNA methylation and mitochondrial DNA copy number, citing existing literature.

Response: We truly value your suggestion. In the third paragraph of the Discussion section (Lines 300-349), we have discussed the mechanisms underlying the evidence obtained from this study, citing a substantial number of previously published papers. However, you are as an expert and experienced researcher in this field, we seek your further guidance. Should there be any additional references or citations that you recommend to substantiate our findings, we would greatly appreciate your guidance on this matter. We look forward to your review and expert evaluation.

Comment 6: the term PhenoAge is not common and needs to be explained.

Response: Thank you for your professional suggestions.The term PhenoAge is not common and needs to be explained.The related context was found in line 103 and as follows: PhenoAge (a DNA methylation predictor trained on 42 clinical markers and age) .

We believe that the revisions we have made have significantly improved the manuscript. We are confident that our responses address all the concerns raised and hope that the revised manuscript meets your expectations.

Thank you once again for your time and effort in reviewing our manuscript. We look forward to your further comments and hope for a positive evaluation.

Sincerely,

Zhijun Feng
South Medical University
fengzhj18@lzu.edu.cn/fengzhj18@sina.com

Round 2

Reviewer 1 Report

Comments and Suggestions for Authors

I appreciated the author's effort in improving the manuscript, but there are still some concerns:

- The assessment that meat consumption is harmful can not be stated in general; I mean that cooking methods or hard processing make meat dangerous, but not meat independently; there is no mechanism by which meat could be harmful. That should be clearly stated.

- Fish consumption implies different lipid profiles (not only protein) so it is impossible to consider with poultry

- The consideration that I asked for probably could not be taken into account, but it should be stated as the limit of the manuscript.

- It is probably incorrect to talk about "healthy meat," but rather, you should talk about processing and cooking.

Comments on the Quality of English Language

Just a revision

Author Response

Dear Professor,

We would like to express our sincere gratitude for your continued engagement and insightful comments on our manuscript. Your expertise and meticulous attention to detail have significantly contributed to enhancing the quality of our work.

We truly appreciate your rigorous and thorough approach in reviewing our manuscript, as well as the comprehensive considerations you have provided. Your suggestions have been instrumental in guiding us to refine our manuscript further.

We are currently addressing each of your comments specifically and will provide a point-by-point response in our revised submission. We are committed to improving our manuscript according to your valuable feedback and hope that our revisions will meet your expectations. We have carefully considered each of your points and have addressed them as follows:

Comment 1: The assessment that meat consumption is harmful can not be stated in general; I mean that cooking methods or hard processing make meat dangerous, but not meat independently; there is no mechanism by which meat could be harmful. That should be clearly stated.

Response: We greatly admire your meticulous and comprehensive approach as a reviewer and are very thankful for the valuable suggestions you provided for our manuscript. Indeed, this issue was overlooked during the writing process. We have revisited the entire text, amended certain descriptions, and highlighted the changes in yellow. Additionally, we have expanded the discussion to include content relevant to this comment, specifically in Lines 278-285: “In summary, this study provides new insights into the relationship between different types of meat intake and aging phenotypes, offering scientific evidence for public health recommendations. It emphasizes the importance of choosing healthy meat products and highlights that the potential harm associated with meat consumption is closely linked to cooking methods and the degree of processing. We advise the public to pay closer attention to meat selection and consumption methods to promote healthy aging. This distinction is crucial in guiding more nuanced and effective public health strategies.“

Thank you once again for your comments, and we look forward to your further review.

Comment 2: Fish consumption implies different lipid profiles (not only protein) so it is impossible to consider with poultry.

Response: We appreciate your expert guidance. During the manuscript preparation, our perspective was somewhat limited, focusing solely on the distinction between red and white meat, while overlooking the differences in their primary components. Taking your feedback into account, we have decided to include this point in the limitations section of our study (Lines 386-391), detailed as follows:

“Fifth, this study is the categorization of both fish and poultry under the same classification of 'white meat.' This approach does not account for the distinct lipid profiles associated with fish consumption, which differ significantly from those of poultry. Fish is rich in omega-3 fatty acids, which have different health implications compared to the predominantly protein-based profile of poultry. This distinction could affect the associations with aging phenotypes observed in our study. “

Thank you once again for your comments, and we look forward to your further review.

Comment 3: The consideration that I asked for probably could not be taken into account, but it should be stated as the limit of the manuscript.

Response: Thank you for your insightful comments and for highlighting important aspects in our manuscript. We sincerely apologize for not fully incorporating your initial suggestions in our previous revision. This oversight was due to our limited understanding at the time, and we regret any confusion caused. We have carefully considered your recent feedback and recognize the importance of addressing the limitation you pointed out. As such, we have revised our manuscript accordingly and explicitly stated this limitation in the discussion section to inform readers. We appreciate your guidance and patience in helping us improve our manuscript, and we hope that these adjustments meet your expectations. Thank you once again.

Comment 4: It is probably incorrect to talk about "healthy meat," but rather, you should talk about processing and cooking.

Response: Thank you for your professional advice. This aspect was indeed overlooked by us. We have incorporated your suggestions into the conclusion section, the abstract, and the final statements of our manuscript, detailed as follows:

Abstract:“This study highlights the significant impact that processing and cooking methods have on meat's role in health and aging, enhancing our understanding of how specific types of meat and their preparation affect the aging process, providing a theoretical basis for dietary strategies aimed at delaying aging and enhancing quality of life.”

Conclusion:

 “These variations highlight the significant impact that processing and cooking methods have on meat's role in health and aging. Consequently, we recommend reducing the intake of processed meats and, where possible, opting for minimally processed alternatives that are cooked in a manner preserving nutritional integrity and reducing harmful effects. This study enhances our understanding of how specific types of meat and their preparation affect the aging process, providing a theoretical basis for dietary strategies aimed at delaying aging and enhancing quality of life. ”

We extend our gratitude once again for your professional, comprehensive, and meticulous feedback. We are resubmitting the manuscript for your review. Thank you again.

Sincerely,

Zhijun Feng
South Medical University
fengzhj18@lzu.edu.cn/fengzhj18@sina.com

Reviewer 3 Report

Comments and Suggestions for Authors

Authors responded all comments which I provided. I have no further comment.

Comments on the Quality of English Language

No

Author Response

Dear Professor,

Thank you for your thoughtful review and constructive comments on our manuscript. We appreciate the time and effort you have invested in evaluating our work.

We are pleased to hear that you are satisfied with the revisions we have made in response to your comments. Your feedback was invaluable in guiding these improvements, and we are grateful for your positive evaluation.

Thank you once again for your contribution to refining our paper.

Best regards,

Zhijun Feng

South Medical University